

# Flavor identification of the stratospheric sudden warmings based on the downward tropospheric influence

Rongzhao Lu[1] and Jian Rao[1*]

[1]Collaborative Innovation Center on Forecast and Evaluation of Meteorological Disasters / Key Laboratory of Meteorological Disaster of Ministry of Education, Nanjing University of Information Science and Technology, Nanjing 210044, China

*Correspondence to*: Jian Rao (raojian@nuist.edu.cn)

**Abstract.** The downward impact of sudden stratospheric warming events (SSWs) on the troposphere is still uncertain. Using the ERA5 reanalysis data, 52 SSWs are identified over the period from 1940–2022, and 33 downward-propagating (DW) SSWs with noticeable impacts on the troposphere are selected with the remaining 19 SSW non-downward-propagating (NDW). The DW events are further classified into three types that are followed by cold surges over the Eurasia (EA), over the North America (NA), and over both (BOTH), respectively. Although the stratospheric polar vortex is weakened and deformed for both DWs and NDWs, the former are stronger and lead to more significant negative Northern Annular Mode (NAM) and North Atlantic Oscillation (NAO) response in the troposphere. For DWs, the anomalous high develops in the polar region, which deflects to lower latitudes, consistent with the frequent appearance of the polar high and the midlatitude blockings. The shape of the anomalous polar high varies with the DWs type, and the extension and deflection of the anomalous high lead to different surface responses. The DWs are also accompanied by a southward shift of the precipitation belt especially over the oceanic and coastal regions. The NDW SSWs show relatively weaker impact on the troposphere, which is primarily related to the weaker amplitude of the stratospheric disturbance. The differences among three types of DWs include diverse NAM structures in the stratosphere, various spatiotemporal evolutions of the NAO pattern in the sea level pressure, different forcing by planetary waves, and varying number ratios between displacement and split. This study reveals the diversity of the DW events and distinguish their potential impact on both continents in the Northern Hemisphere.

## 1 Introduction

In the northern winter stratosphere, the polar temperature increases dramatically in just a few days when



the polar vortex deforms or even collapses and the circumpolar westerly winds decrease suddenly and even reverse the direction (Baldwin et al., 2021). This phenomenon, known as stratospheric sudden warmings (SSWs), is an important manifestation of stratosphere-troposphere coupling and is most

common in spring and winter. The occurrence of SSW is associated with strong, upward-propagating planetary waves from the troposphere (Sjoberg and Birner, 2012;Butler et al., 2015). Recent studies also found the stratospheric precondition might play a decisive role in inducing the SSW event, which determine the intensity of the interaction between planetary waves and the mean flow. Butler et al., (2015) suggested that SSW is caused by a sharp increase in the amplitude of planetary-scale waves (primarily

wave 1 and 2) propagating upward from the troposphere and perturbing the stratospheric polar vortex. Modeling evidence shows that the wave-1 forces the displacement of the polar vortex from the North Pole (Lindgren and Sheshadri, 2020), while the wave-2 determines the extent of vortex elongation and deformation (Baldwin et al., 2021).

The stratospheric variability associated with SSWs usually lead the tropospheric variation through the

stratosphere-troposphere coupling (Hitchcock and Simpson, 2014; Wu and Reichler, 2019). Several mechanism have been proposed to explain the possible downward impact of the stratosphere on the troposphere, including wave-flow interaction theory (Kuroda and Kodera, 1999), balanced flow dynamics theory (Black, 2002; Haynes, 1991), planetary wave refraction theory (Schmitz and Grieger, 1980), non-local potential vorticity response (Ambaum and Hoskins, 2002), and isentropic atmospheric

meridional mass circulation theory (Cai and Shin, 2014). The wave-flow interaction theory suggests that upward-propagating tropospheric wave forcing fluctuations affect stratospheric mean flows, and changes in stratospheric background flows in turn affect the vertical propagation of planetary waves (Kuroda and Kodera, 1999; Hartmann, 2000). As a consequence, the stratospheric disturbances cause significant atmospheric circulation changes and near surface climate anomalies (Colucci and Kelleher, 2015;

Dall'Amico et al., 2010).

SSWs can have a considerable impact on the troposphere and have a sustained effect on surface weather for weeks or even months (Domeisen et al., 2020; Rao et al., 2021; Lu et al., 2023). The weakened polar vortex during SSW is usually projected onto the negative phases of the Northern Annular Mode (NAM) and/or the North Atlantic Oscillation (NAO), which gradually propagate downward (Karpechko et al.,

2017; Kunz and Greatbatch, 2013; White et al., 2019). Changes in the NAM are often accompanied by equatorward shifts in storm paths and tropospheric jets (Kidston et al., 2015), shifts in the centre of the



East Asian jets, variation in the blocking frequency (Anstey et al., 2013), increased possibility of cold air outbreaks over Eurasia and North America (Baldwin and Dunkerton, 2001; Lehtonen and Karpechko, 2016; Lu et al., 2022; Yan et al., 2022), and populated likelihood of extreme extreme rainfall (Karpechko

and Manzini, 2012).

The common characteristics of SSWs have been widely reported in literature (see Baldwin et al., 2021 and references therein). However, every SSW has its individual features and displays strong particularity (Karpekho et al., 2018; Rao et al., 2018, 2019, 2021; Lu et al., 2023). It is the differences between individual SSWs that distinguish their influence on the troposphere, which varies in extent degree, area

and scope. Possible factors explaining the SSW individuality especially in its influence include the SSW strength, the initial warming location or the warming center (Zhang and Chen, 2019; Yan et al., 2022), and the warming duration time (Hitchcock and Simpson, 2014), details of the wave flux between the troposphere and stratosphere (Shi et al., 2024), and the geometry of the polar vortex (split or displacement) (Maycock and Hitchcock, 2015; Rao et al., 2020). Therefore, the SSWs can be classified into different

types based on those metrics.

Different SSW classifications are widely used in literature. For example, based on vortex geometry shape the SSWs are classified as split, displacement, and mixed types (Charlton and Polvani, 2007; Rao et al., 2019). Based on whether the stratosphere reflects planetary waves during the westerly recovery phase following the SSW onset, it can be divided into absorbing and reflecting types (Kodera et al., 2016).

According to whether the event has had a significant impact on the troposphere, it can be grouped into downward (DW) or non-downward (NDW) types (Jucker, 2016; Runde et al., 2016; Karpechko et al., 2017; White et al., 2019). Although the DW SSWs show a dipping NAM signals from the troposphere to the stratosphere, the near surface response structure is still different among DWs, with cold extreme sometimes only appearing in Eurasia, sometimes only in North America, and sometimes in both

continents. However, there is still not a widely accepted subclassification for DWs events based on the coverage of the near-surface cold anomalies associated with SSW. This study is mainly concerned with two questions: (1) What causes the inter-case difference in the DW influence on the troposphere although the NAM during DWs shows downward propagation to the lower troposphere? (2) What can we learn from the flavor identification for DWs.

The organization of the paper is constructed as follows. Following the introduction, Section 2 describes the data and methods used in this paper. Section 3 compares the tropospheric response characteristics of





various DW events. Section 4 analyzes the dynamics related to various DW events. Finally, Section 5 provides a summary and discussion.

## 2 Data and methods

### 2.1 Reanalysis data

We use the European Centre for Medium-Range Weather Forecasts (ECMWF) fifth generation reanalysis (ERA5) dataset over the period from October 1940 to December 2022. Both three-dimensional and two-dimensional data are used. The three-dimensional variables on the isobaric levels include the air temperature, the geopotential, the zonal and meridional winds, and the vertical velocity on p coordinate.

The two-dimensional variables on single levels include the total precipitation and the temperature at two meters. The isobaric levels vertical levels ranging from 850 hPa to 1 hPa, as well as surface data. The horizontal resolution of the data is 2° latitude by 2.5° longitude.

### 2.2 Methods

#### 2.2.1 DW and NDW event definitions

We use the WMO definition (Andrews et al., 1985) to identify SSW. Specifically, All days when a change in the zonal-mean zonal wind $\bar{u}$ from westerlies to easterlies occurs at 60°N and 10 hPa within the period from the 1 November to 31 March each year are selected. The first day when $\bar{u}$ undergoes a transition from westerlies to easterlies is defined as the onset date of this SSW. Several SSW definitions considering the wind reversal at different latitudes and heights in the circumpolar region have been

compared by Butler and Gerber (2018). They concluded that a small modification in the wind position or the wind threshold using for the SSW definition do not lead to major changes in the statistics of the SSWs. Considering the zonal winds change radically even after the SSW onset, the onset date of the two SSW events must be more than 20 days apart to avoid counting twice for the SSW with zonal winds oscillating between westerlies and easterlies. Further, $\bar{u}$ must have recovered to westerlies at least for

10 consecutive days prior to 30 April, which can exclude final warmings (e.g., White et al., 2019). Finally, 52 SSWs were selected, and the average occurrence frequency of SSW is approximately 0.63 per year, consistent with previous studies (Liu et al., 2019; Rao et al., 2021ERL).

To classify the SSW as the DW or the NDW, the NAM index is extracted using the reanalysis (Baldwin and Thompson, 2009). After removing the daily climatological mean from the geopotential heights, the



anomaly data are area-weighted (i.e., multiplied by the cosine of the latitude) over 20˚ -90°N. The

anomaly data are normalized for each pressure level, and the empirical orthogonal function is performed

to extract the leading mode (i.e., the NAM pattern). The anomaly field is projected onto the leading mode

to compute the corresponding timeseries (i.e., the NAM index). The NAM index for each pressure level

is calculated separately, and the NAM at 150 and 1000 hPa are used to identify DWs.

The definition of DWs used in this paper follows the method by Natarajan et al., (2019). Namely, a DW

is defined when the SSW satisfies the following three conditions: During the 45-day period from day 8

to day 52 relative to the SSW onset date, 1) the average NAM index is negative; 2) the percentage of

days with a negative NAM index is greater than 70% at 150 hPa; and 3) the percentage of days with a

negative NAM index is greater than 50% at 1000 hPa. To reduce the effect of topographic complexity

over high-terrain regions, White et al., (2019) adjusted 1000 hPa to 850 hPa for the third condition, which

is also included in this study. Based on these criteria, 33 DWs are identified, which account for 63% of

all SSWs.

### 2.2.2 Classification of DWs

It has been revealed from the composite for all SSWs that continental cold anomalies usually develop

over Eurasia and North America, implying an increase in cold air outbreaks after the DW SSW onset. To

better describe and distinguish the DWs, we further divide DWs based on the inland temperature

anomalies within 40 days after the onset of DWs. Considering that cold surges are more active over the

Eurasia (Europe + Asia) and North America (US + Canada) following the DWs, a comparison between

the cold anomalies over the two regions can further classify the DWs into three types: cold anomalies

only appearing over North American type (NA), only appearing over Eurasia (EA), and over both regions

(BOTH). Regions focused in this study include Europe (40°-70°N, 0°-60°E), Asia (40°-70°N, 60°-

140°E), United States (30°-46°N, 70°-120°W), and Canada (46°-60°N, 60°-120°W). Within one region,

the area-averaged temperature anomalies over a 40-day period are considered to be associated with the

downward impact of the DW if the anomalies meet the following two criteria: 1) the percentage of days

with cold anomalies is greater than 50%; 2) the mean temperature anomaly over a 40-day period is less

than -0.3°C.

SSWs classified as split and displacement events is based on the method by Esler et al., (2009) using the

two-dimensional (2D) moment analysis method. Maycock and Hitchcock (2015) modified this analysis



method by using the geopotential height at 10 hPa instead of the potential vorticity used in Esler et al.,

(2009). Several modifications for some parameters in this study are as follows. The first is the edge of

the polar vortex, which we define as the climatological mean geopotential height over 60°N and 10 hPa

in winter (e.g., Maycock and Hitchcock 2015). The second is the thresholds for the split and displacement

SSWs. We choose the thresholds as the most equatorward 5.7% of centroid latitudes and largest 5.2% of

aspect ratios, yielding a threshold of 62.9°N for centroid latitude and 2.46 for aspect ratio, respectively

(e.g., Seviour et al., 2013).

**Table 1**. Statistic of the SSWs from 1940–2022. The second column (S/D) shows the SSW type based on the vortex shape (D=displacement, S=split). The third column shows the SSW type based on whether the stratospheric signal propagates downward (DW=downward, NDW=non-downward). The last column shows the subclassification of DWs.

| onset date | S/D | DW/NDW | type | onset date | S/D | DW/NDW | type |
|---|---|---|---|---|---|---|---|
| 1941-02-07 | D | DW | BOTH | 1981-03-04 | D | NDW | — |
| 1942-03-21 | D | NDW | — | 1981-12-04 | D | NDW | — |
| 1945-03-19 | S | NDW | — | 1984-02-24 | D | DW | BOTH |
| 1946-02-19 | D | NDW | — | 1985-01-01 | S | DW | BOTH |
| 1946-03-19 | D | NDW | — | 1987-01-23 | D | DW | EA |
| 1950-03-05 | D | NDW | — | 1987-12-08 | S | NDW | — |
| 1952-02-22 | S | DW | BOTH | 1988-03-14 | S | NDW | — |
| 1952-11-19 | D | DW | EA | 1989-02-21 | S | NDW | — |
| 1954-12-18 | D | DW | EA | 1998-12-15 | D | NDW | — |
| 1955-01-26 | S | DW | BOTH | 1999-02-26 | S | DW | EA |
| 1957-02-04 | S | DW | EA | 2000-03-20 | D | NDW | — |
| 1958-02-01 | D | DW | NA | 2001-02-11 | S | DW | NA |
| 1960-01-17 | D | DW | BOTH | 2001-12-30 | D | NDW | — |
| 1963-01-27 | S | NDW | — | 2002-02-17 | D | DW | NA |
| 1965-12-16 | D | NDW | — | 2003-01-18 | S | NDW | — |
| 1966-02-22 | S | DW | EA | 2004-01-05 | D | DW | NA |
| 1968-01-07 | S | DW | EA | 2006-01-21 | D | DW | EA |
| 1968-11-28 | D | DW | BOTH | 2007-02-24 | D | NDW | — |
| 1969-03-13 | D | NDW | — | 2008-02-22 | D | DW | NA |
| 1970-01-02 | D | DW | BOTH | 2009-01-24 | S | DW | EA |
| 1971-01-18 | S | DW | BOTH | 2010-02-09 | S | DW | EA |
| 1971-03-20 | D | DW | BOTH | 2010-03-24 | D | DW | EA |
| 1973-01-31 | S | NDW | — | 2013-01-06 | S | DW | EA |
| 1977-01-09 | S | DW | BOTH | 2018-02-12 | S | DW | EA |
| 1979-02-22 | S | DW | BOTH | 2019-01-01 | S | DW | NA |
| 1980-02-29 | D | DW | BOTH | 2021-01-05 | D | DW | EA |



**2.2.3 isentropic potential vorticity**

In this paper, isentropic potential vorticity (IPV) is used. The vertical component of IPV is defined (Hoskins et al., 1985) as:

$$IPV = -\frac{g\left(f + \vec{k} \cdot \nabla_\theta \times \vec{V}\right)}{\frac{\partial p}{\partial \theta}}, \tag{1}$$

where $f$ is the planetary vorticity and $\theta$ is the potential temperature.

**2.2.4 E-P flux**

The E-P flux and its divergence can characterize the propagation of quasi-geostrophic planetary waves and its interaction with the mean flows. The E-P flux and its divergence are employed to diagnose the dynamical processes during SSWs as follows (Edmon et al., 1980):

$$F_\varphi = -a(cos\,\varphi)\overline{u'v'}, \tag{2}$$

$$F_p = a(cos\,\varphi)f\frac{\overline{v'\theta'}}{\overline{\theta_p}}, \tag{3}$$

$$\nabla \cdot \boldsymbol{F} = \frac{1}{a\,cos\,\varphi}\frac{\partial\left(F_\varphi\,cos\,\varphi\right)}{\partial\varphi} + \frac{\partial F_p}{\partial p}, \tag{4}$$

where $F_\varphi$ is the horizontal component of the E-P flux, $F_p$ is the vertical component, and $\nabla \cdot \boldsymbol{F}$ is the divergence of the E-P flux, and $a$, $\varphi$ are the Earth's radius and latitude, respectively. The E-P flux vector characterize the propagation direction of the planetary waves. The E-P flux divergence indicates the effect of the planetary waves on the mean flow. When the E-P flux is convergent (divergent), it indicates that there is easterly (westerly) forcing of the mean flow.

**3 Comparison of tropospheric responses to three types of DWs**

**3.1 Spatiotemporal evolution of the NAM**

The NAM index is a substitute index that can be used to describe the downward propagation of stratospheric disturbance for DWs. The composite evolutions of the NAM index for the three types of DWs are compared in Fig. 1. Consistent with previous study (Karpechko et al., 2017; Kunz and Greatbatch, 2013; White et al., 2019), radical negative NAM signals appear above 200 hPa within a few days around the SSW onset date. After the DW onset, the negative signal propagates downward into the troposphere (Fig. 1a-c), forming a typical dripping-paint pattern in the troposphere (e.g., Baldwin and Dunkerton 2001). On day 20 and afterward, the upper stratosphere gradually recovers to the positive



NAM, while the negative signal below 50 hPa return to the positive NAM at different times for the three

types of DWs. The negative NAM signal below 50 hPa can persist until day 60 and even afterward,

suggesting that the downward influence of DWs are persistent, providing a possible predictability source

for the troposphere (Rao et al., 2021; Lu et al., 2023). Differences in the NAM evolution are noticed for

190     the three types of DWs.

1) In the pre-SSW onset period, the NAM displays different behaviors. The positive NAM mainly

develops around day -50 and day -30 with the most significant signals in the lower stratosphere and upper

troposphere for the type BOTH. The positive NAM only develops in the upper stratosphere around day

-30 for the type EA. In contrast, the positive NAM dramatically develops from day -50 in the upper

195     stratosphere to day -15 in the lower troposphere, displaying a noticeable downward propagation.

2) In the post-SSW onset period, the negative NAM is structured in different spatiotemporal dripping

shape. Around 20 days or more after the SSW onset, the NAM is reversed in the upper stratosphere, while

the reversion time in lower levels is later. The negative NAM at 50 hPa persist longer than any other level

for all types of DWs and even NDW. In contrast, the NAM sign reversion is earliest for NA out of the

200     three DW types around day 50 at 50 hPa (Fig. 1c), while the NAM sign change is much slower for BOTH

and EA beyond day 60 (Fig. 1a, b). It is also seen that the NAM sign change for NDW is also around day

50 at 50 hPa (Fig. 1d).

3) The positive NAM intensity before the SSW onset and the negative NAM intensity after the SSW

onset are variously contrasted for the DWs and NDW. The NAM evolves from moderately positive to

moderately negative for the type BOTH. It evolves from weakly positive to weakly but persistently

negative for type EA. Further, it evolves from intensely positive to intensely but shortly negative for type

NA. The positive NAM develops 40 days or more before the NDW, but the negative NAM intensity and

the NAM contrast before and after the NDW onset are weak.

4) The near surface exhibits different behaviors in the NAM intermittent signals for the three types of

DWs. Specifically, the negative NAM at the near surface is continuous for BOTH and EA types, while it

is very short in the persistent time for NA. At the very beginning of the NA DWs, the NAM is still

positive at the near surface due to the lagged downward impact of the stratosphere, with the negative

NAM from day 10 to day 50. In contrast, the negative NAM signal fails to appear at the near surface for

NDWs, only with significant positive NAM around day 30.





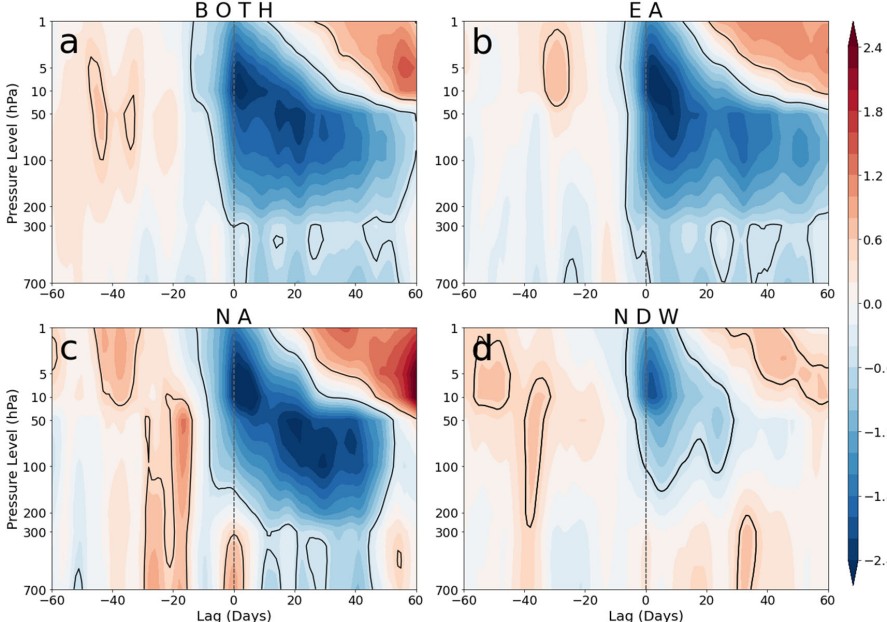

**Figure 1.** The composite evolution of the NAM index for (a-c) three types of DWs and (d) NDW events. (a) BOTH, (b) EA, (c) NA and (d) NDW events. The units are in standard deviations. The black line represents statistical significance at the 95% level.

The NAO index is closely related to low-level circulation, which are significantly correlated with the probability of extremes, such as regional coldness, snowstorm, and strong winds (Thompson and Wallace, 2001; Scaife et.al, 2014). Figure 2 compares the probability density functions (PDFs) of the mean NAO index over the 60 days after the onset for each type of event. Comparisons with a random sample of 2000 winters show that the probability of the NAO index for DWs shifts left toward negative, which indicates

that DWs have a significant effect on the surface circulation by modifying the PDF of the NAO. The NAO index after the DW onset for BOTH is 2-3 times more likely to be less than -0.8 than for other cases, with the median roughly located at -0.7, and the probability that the index is positive is almost zero. The median values of the NAO after EA and NA are the same, roughly located at -0.5, but the difference is noticeable. Namely, the PDF of the NAO for NA is more dispersed and the PDF peak is

smaller than that for EA and BOTH. In contrast, the type BOTH showed the largest composite mean of NAM (-0.762), followed by EA (mean NAM = -0.567) and NA (mean NAM=-0.435). The PDF of NAM for NDW is primarily concentrated between -0.2 and 0.4 (68.4%), with median and mean values near 0.1 and 0.088, which might indicate that the NAO nearly has no preference during NDW events.




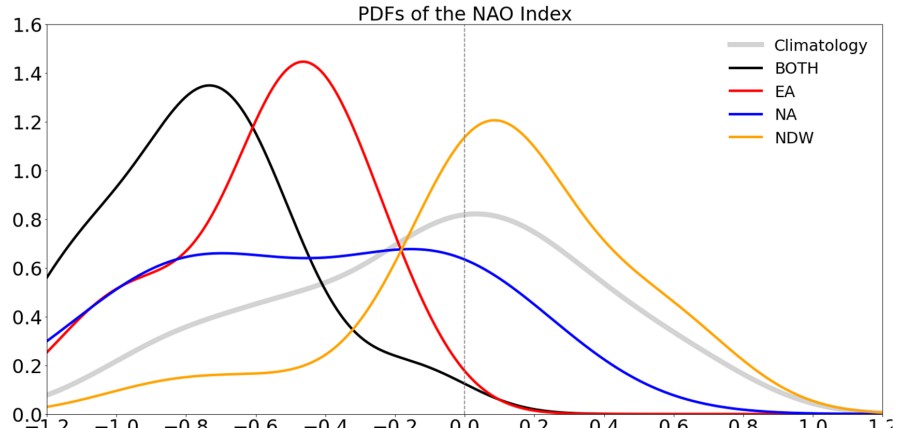

**Figure 2.** Probability density functions for the normalized daily NAO index during 2000 randomly averaged 60-day periods in winter (grey curve), the 60 days after onset data of BOTH (black curve), EA (red curve), NA (blue curve), and NDW (orange curve) events.

### 3.2 Comparison of near surface response

230  To compare the possible different impact of DWs on the near surface, the composite t2m anomalies in the 40-day interval before and after the SSW onset is shown in Fig. 3. The cold anomalies appear over different region during the SSW for DWs and the NDW. Specifically, in the pre-SSW period (day -40 to day 0), significant cold anomalies have well developed over northern Eurasia for BOTH and EA, while the cold anomalies are not detectable over Eurasia for NA (Fig. 3a-c). Although cold anomalies also

235 appear over Eurasia, the anomaly magnitude is fairly weak for NDWs (Fig. 3d). The western coast of North America is covered with cold anomalies for BOTH and NDW (Fig. 3a, d), while significant warm anomalies appear over most of North America for EA and NA (Fig. 3b, c). It is also found that significant warm anomalies develop over Europe for EA (Fig. 3c).

  In the post-SSW period, the t2m anomaly pattern is contrastingly different for the three types of DWs

240 and the NDW. For the type BOTH, the cold anomalies in the Eastern Hemisphere continue to strengthen and expand southward, covering most of Eurasia; the cold anomalies in the Western Hemisphere expand from western coasts of North America to most of the continent (Fig. 3e). For the type EA, the cold anomalies persist over northern Eurasia and the cold center move further eastward to Northeast Asia, while the warm anomalies over eastern Canada and southern Europe strengthen (Fig. 3f). For the type

245 EA, warm anomalies over western Europe also intensify, while the warm anomalies over North America



are replaced by anomalous coldness with the center over the central US (Fig. 3g). For the NDW, the t2m anomalies over both lands are very scattered and less organized, although patches of warm anomalies are observed over Asia (Fig. 3h).

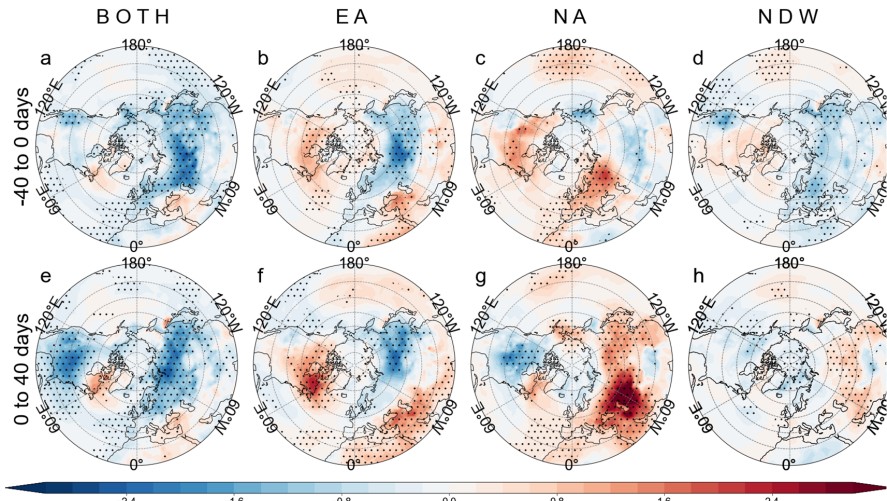

**Figure 3.** Composite 2-meter temperature (t2m) anomalies (shadings; units: K) for (a, e) BOTH, (b, f) EA, (c, g) NA and (d, h) NDW events before the SSW onset (top row) and afterward (bottom row). The composite is based on the mean of 40-day intervals. The dots mark the composite anomalies at the 95% confidence level using the *t*-test.

To well depict the dynamic process of the near surface response to the three types of NDs, the composite

evolution of t2m anomalies over Eurasia and North America is shown in Fig. 4. For the type BOTH, cold anomalies develop throughout the SSW occurrence period from day -40 to day 40 over all the four regions (Europe, Asia, US, and Canada), except that the negative t2m anomalies at Canada are relatively weak and exhibit relatively large subseasonal variability (Fig. 4a). For the type EA, cold anomalies are persistent over northern Eurasia, larger in Asia than in Europe, while warm anomalies are persistent over

North America, larger in Canada than in US (Fig. 4b). For the type NA, the reversal of temperature anomaly sign is observed over these four regions. The reversal of temperature anomalies from positive to negative indicate outbreaks of cold air (Lehtonen and Karpechko, 2016). Namely, cold air outbreaks increase in North America, while the anomalously cold state gradually recovers to normal in Asia with stable moderately warm state in Europe (Fig. 4c). For NDWs, the temperature anomalies are fairly weak

throughout the SSW onset except that Canada experiences a transition from anomalously warm state to cold (Fig. 4d).



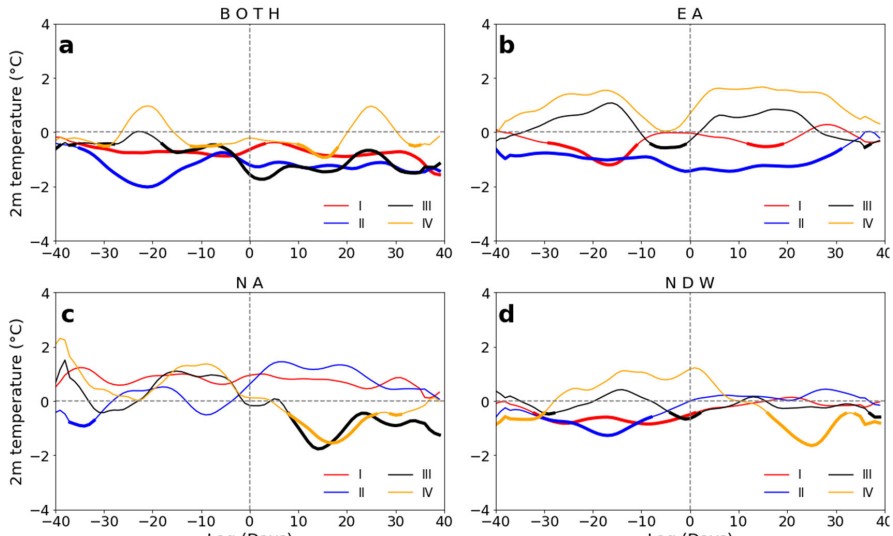

**Figure 4.** Evolution of the area-averaged t2m anomalies (units: K) from day -40 to day 40 over four regions (I–IV), including Europe (40°-70°N, 0°-60°E), Asia (40°-70°N, 60°-140°E), United States (30°-46°N, 70°-120°W), and Canada (46°-60°N, 60°-120°W) for (a) BOTH; (b) EA; (c) NA and (d) NDW events. Colored lines denote the four areas from I to IV, respectively. The thickened part of the line denotes the composite at the 95% confidence level.

### 3.3 Analysis of isentropic potential vorticity

In order to further explore the process of cold air activity, an analysis on the isentropic potential vorticity (IPV) is investigated for the three types of DWs and the NDW. According to Eq. (1), the IPV is

proportional the absolute vorticity and static stability. In the case of equal vorticity, the cold air mass usually has a higher potential vorticity value due to its relatively larger static stability, which can be used to track the cold air activity on the isentropic surface (Hoskins et al., 1985; Lu and Ding, 2015;). The 315K is-entropic level has been used to track the cold air sources by detecting the movement of high IPV center (Jeong et al., 2006). The composite evolutions of IPV anomalies are shown in Fig. 5 for all types

of events. For the type BOTH, a patch of high IPVs appeared from North Pacific 30-20 days prior to the event and gradually move southeastwards in the following period, which finally reach the North America (Fig. 5a). During day 10 to 20, anomalously high IPV air moves to 45°N and even more southward, with the anomaly amplitude gradually weakened. In contrast, there are two high IPV centers in Eurasia, one over North Atlantic, and the other over Central Asia. The anomalously high IPV center over North

Atlantic gradually moves to Europe, indicating the cold air outbreak. The high IPV center over Central



Asia is more stable from day -30 to -10 and redevelops from day 0 to 20.

For the EA events, the high IPV center in the North Atlantic is weak and significant high IPV anomalies appear in Europe (Fig. 5b). The high IPV anomalies developed in North Atlantic soon after the SSW onset, and the Arctic is nearly covered by the positive IPV anomalies. Another positive IPV anomaly

center is detected over North Asia since day -30 to -20, which diminish from day -20 to -10 and redevelop from day -10 to 10. The high IPV patch moves southeastward to East Asia after day 10, indicating cold air outbreaks in local regions.

For the NA events, the high IPV center first appears over North Pacific from day -20 to -10, which is still active from day -10 to 10 (Fig. 5c). The positive IPV anomaly intensity weakens from day 0 to 10 and

then redevelops in the later period. Further, it is also noticed that a large patch of positive IPV anomalies form over North Atlantic after the SSW onset, which is nearly motionless.

For the NDW events, the IPV anomalies are relatively weak, and the positive IPV anomalies are scattered and less organized over the land (Fig. 5d), consistent with the distribution of t2m anomalies. Weak positive IPV anomalies primarily appear over the oceans and are nearly motionless. It is also observed

that a narrow band of positive IPV anomalies exist from the west to east across Canada during day 20 to 30.



**Figure 5.** Composite isentropic potential vorticity (IPV) anomalies (shadings; units: PVU, 1 PVU = $10^{-6}$ m$^2$ K kg$^{-1}$) at 315 K for (a) BOTH, (b) EA, (c) NA, and (d) NDW events. The composite is based on the mean of 10-day intervals. The dots mark the composite anomalies at the 95% confidence level using the *t*-test.



### 3.4 Total precipitation

Previous studies have shown that SSWs not only influence the cold air outbreaks, but also affect the rainfall anomalies (King et al., 2019; Oehrlein et al., 2021). Negative NAM phases are usually

accompanied by a possible equatorward shift of the storm track, and a possible intensification of the moving cyclone at low latitudes (Afargan-Gerstman and Domeisen, 2020; McAfee and Russell, 2008; Thompson and Wallace, 2001). The composite precipitation anomalies from day 0 to 40 are shown in Fig. 6 to compare the rainfall anomaly sensitivity to the SSW type. The most pronounced common features for the four conditions are the rainfall dipole structure over North Atlantic - Europe. Positive

rainfall anomalies form over Atlantic midlatitudes and the Mediterranean Sea, while negative rainfall anomalies form over Atlantic high-latitudes. This rainfall dipole is strong for BOTH and EA, while the intensity is relatively weak for NA and NDW. The positive rainfall anomaly in midlatitudes for NA is broken into two chains, one over the ocean, and the other biased toward the Eurasian land (Fig. 6c). Although the NAM fails to descend to the near surface, the rainfall dipole is still present for NDW (Fig.

6d).

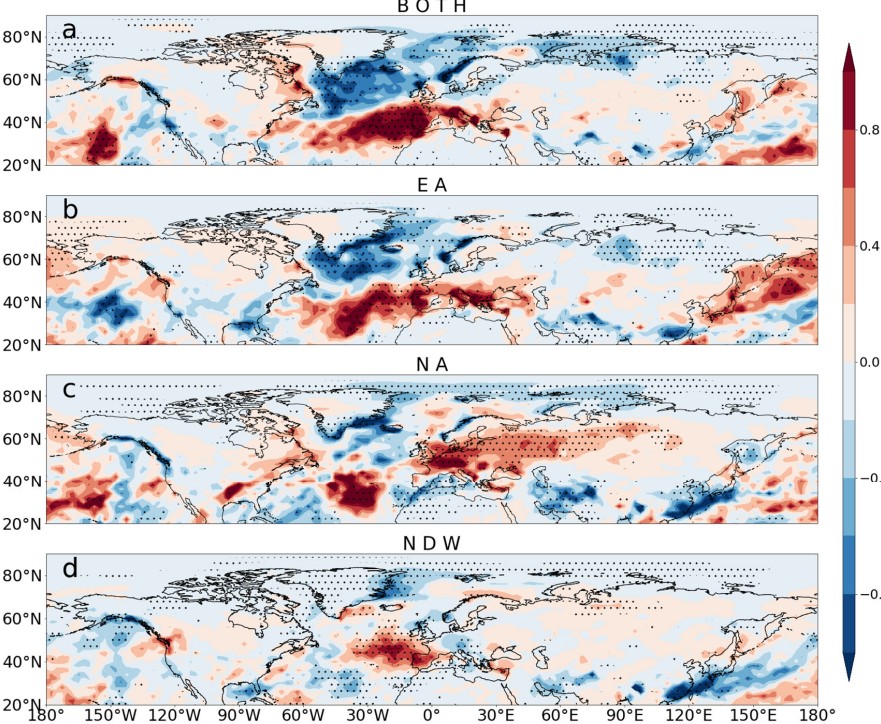

**Figure 6.** Composite precipitation anomalies (shadings; units: mm day⁻¹) for (a) BOTH, (b) EA, (c)



NA and (d) NDW events after the SSW onset. The composite is based on the mean of day 0 to 40. The dots mark the composite anomalies at the 95% confidence level using the *t*-test.

## 4 Dynamic diagnostics

In order to compare the large-scale atmospheric dynamics between different types of SSW events and to better understand their downward influences, the composite circulation anomalies at 100 hPa and the sea level pressure anomalies are shown in Fig. 7. In the pre-SSW period, the circulation structured is different

organized for three types of DWs and the NDW. For the type BOTH, an anomalous high appears over Canada, and a low anomaly center forms over central northern Eurasia (Fig. 7a), implying the displacement of the polar vortex toward North Asia. On the near surface, positive MSLP anomalies prevail over the Arctic, and negative anomalies develop over midlatitudes. The negative NAM pattern has well developed before the SSW onset for the type BOTH. The height anomaly distribution at 100

hPa for the type EA in the pre-SSW period is similar to that for BOTH except that the height anomaly centers are further eastward situated (Fig. 7b). The anomalous high is situated over Europe, while the anomalous low is situated over Northeastern Asia. Positive MSLP anomalies develop over the Arctic and northern Europe, while the negative MSLP anomalies form over northern Canada. For the type NA, the preceding circulation anomalies especially at the near surface exhibit different patterns (Fig. 7c). The

anomalous high at 100 hPa is centered over the lakes, while the low covers most of northeast Asia. However, the near surface is covered by the anomalous low over most of the Arctic. For NDWs, the polar vortex at 100 hPa is not significantly disturbed, although the near surface exhibits a pressure anomaly dipole, with the low over the Bering Strait – Alaska and the high over the northern Europe (Fig. 7d). Comparing the four types of events, the precursor circulation anomalies for BOTH and EA are more

baroclinic from the lower stratosphere to the troposphere (Fig. 7a, b), while for NA and NWDs, the circulation anomalies are nearly barotropic in high latitudes (Fig. 7c, d).

In the post-SSW period, the Arctic is completely covered by high anomalies from the near surface to the lower stratosphere with a nearly barotropic circulation structure (Fig. 7e-h). In the lower stratosphere, the Arctic for all types of events is covered by the anomalous high, indicating the breakup of the

stratospheric polar vortex. The positive anomaly amplitude for the three types of DWs are comparable, and their differences are mainly featured by midlatitude circulation anomalies. For BOTH, negative height anomalies are clearly present in midlatitudes with three centers, one over North Atlantic, one over



Canada, and one over East Asia (Fig. 7e). Similarly, the MSLP anomalies are well structured in a negative

NAM pattern with the negative anomalies maximized over North Atlantic along midlatitudes. For EA,

the negative height anomalies are not as detectable as for BOTH, and only the negative center over East

Asia is clearly observed (Fig. 7f). On the near surface, the NAM structure is more clearly present in the

Atlantic sector than in the Pacific sector. The local anomalous high over North Pacific cut the annular

structure in the Eastern Hemisphere. For NA, the anticyclonic anomalies at 100 hPa are further biased

toward the Arctic Canada, while nominal negative height anomaly band in midlatitudes for the negative

NAM is only present over North Pacific (Fig. 7g). Similarly, the negative MSLP anomaly band is

concentrated from the Western Europe to East Asia and North Pacific (Fig. 7g), different from the long

anomalous low band in midlatitude for BOTH and EA. For NDWs, although positive height anomalies

are present over the Arctic in the stratosphere (Fig. 7h), the amplitude is nearly half or one third of the

strength for DWs. The MSLP anomalies in Arctic are even insignificantly detectable, while significantly

negative MSPL anomalies over North Atlantic–Northern Eurasian are observed.

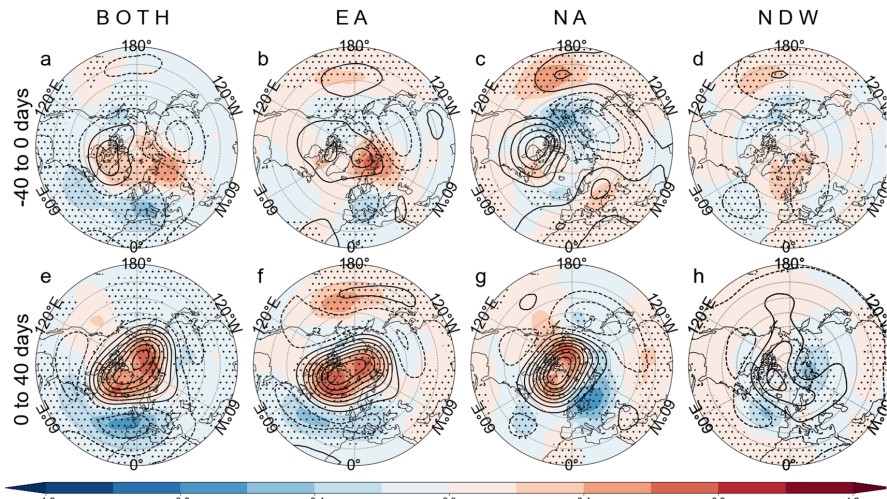

**Figure 7.** Composite 100-hPa geopotential height anomalies (contours; units: gpm) and sea level
pressure anomalies (shadings; units: hPa) for (a, e) BOTH, (b, f) EA, (c, g) NA and (d, h) NDW events
before the SSW onset (top row) and afterward (bottom row). The composite is based on the mean of
40-day intervals. The dots mark the composite anomalies at the 95% confidence level using the *t*-test.

To further analyze the wave dynamics and to better understand the difference between the three types of

DWs and the NDW, the E-P flux anomalies and the E-P flux divergence anomalies are shown in Fig. 8.

The ten-day intervals before the SSW onset and afterward are examined for total planetary waves. For



all types of SSWs, the upward propagation of planetary waves is enhanced before the event onset (Fig.

8a-d). The enhancement of the upward-propagating waves before DWs are nearly twice as strong as that

for the NDW. Comparing the three types of DWs, the E-P flux anomalies are very comparable, and the

anomalous E-P flux convergence center is structured differently (Fig. 8a-c). It is revealed that the EP flux

convergence is strongest for the type BOTH and weakest for the type NA. The E-P flux convergence

anomalies are weaker for the NDW than all types of DWs, likely due to the weak E-P flux itself.

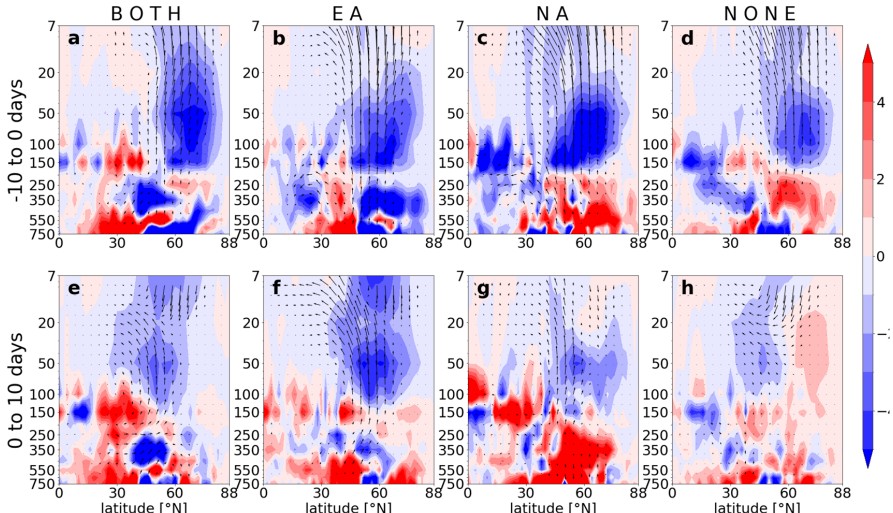

**Figure 8.** Composite E-P flux anomalies (vectors; units: $F_y$ in $10^4$ kg$^3$ s$^{-2}$, $F_z$ in $10^6$ kg$^3$ s$^{-2}$) and E-P flux divergence anomalies (shadings; units: m s$^{-1}$ d$^{-1}$) for (a, e) BOTH, (b, f) EA, (c, g) NA and (d, h) NDW events before the SSW onset (top row) and afterward (bottom row). The composite is based on the mean of 10-day intervals.

After the SSW onset, the composite 10-day interval E-P flux anomalies are also not identical for the three

types of SSWs (Fig. 8e-g). The upward propagation in the lower stratosphere and troposphere is still

enhanced for all types of DWs, while it begins to weaken in the extratropical upper stratosphere. As a

consequence, the anomalous E-P flux convergence is still present in the lower stratosphere for all types

of DWs. In contrast, weakening of the anomalous E-P flux convergence begins to form for the NDW,

and the anomalous downward propagation of waves is also stronger for the NDW than the DW (Fig. 8h).

The relatively short lifetime of the wave forcing for the NDW likely explains the relatively weak intensity

of the stratospheric disturbance and even lacking impact on the near surface in the later period. It is also

revealed that the E-P flux divergence (or convergence) anomalies in the upper stratosphere are much

weaker than in the lower stratosphere and upper troposphere all for events (Fig. 8e-h).



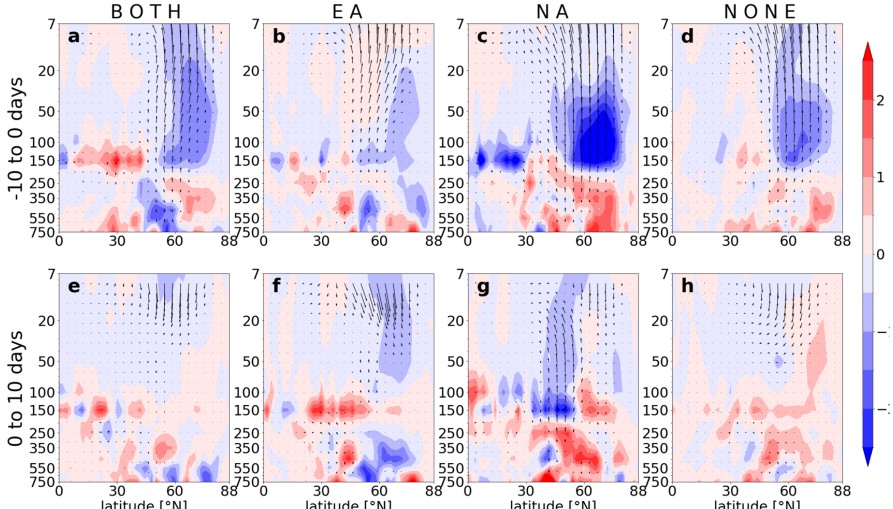

**Figure 9.** Same as in Figure 8, but for planetary wave-1 E-P flux.

Only planetary waves can propagate into the stratosphere, and the contribution of the wave 1 and wave 2 to the total E-P flux and its divergence is shown in Fig. 9 and Fig. 10, respectively. A decomposition of the waves for E-P flux before the SSW onset can easily reveals the difference among the three types of DWs. Specifically, in the pre-SSW onset, the upward propagation of planetary wave 1 is strengthened for all events, and the E-P flux convergence anomalies by wave 1 also appear for all events (Fig. 9a-d).

Although the near surface impact only forms over North America for the NA events, the E-P flux convergence anomalies by wave 1 are strongest for NA out of all groups (Fig. 9c).

In the post-SSW 10-day interval period, the upward propagation of wave 1 is suppressed for all events, and the contribution of wave 1 to the deceleration of westerlies (and therefore weakening of the polar vortex) nearly disappears (Fig. 9e-h). Namely, the E-P flux anomalies change the direction from upward

to downward, and the E-P flux divergence anomalies are nearly zero. Wave forcing in midlatitudes are still present for DWs, and the anomalous E-P flux convergence also persists in the lower stratosphere at midlatitudes (Fig. 9e-g).



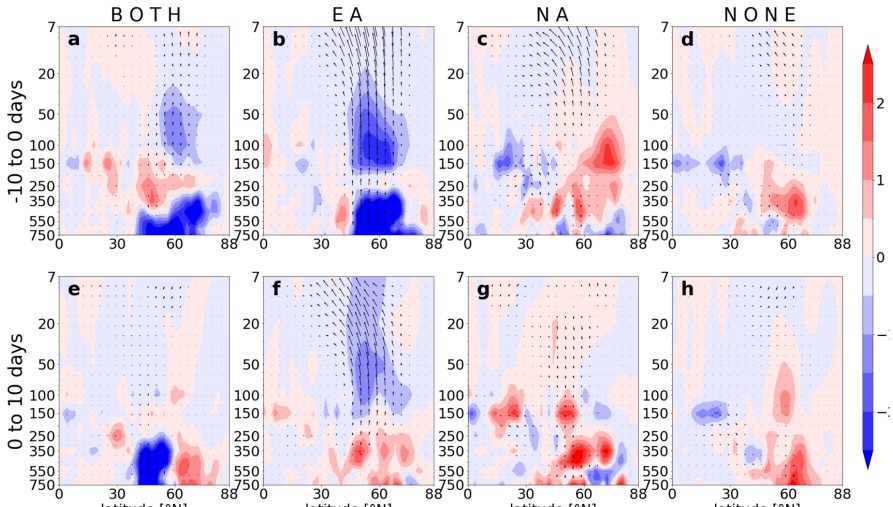

**Figure 10.** Same as in Figure 8, but for planetary wave-2 E-P flux.

The dynamics difference for the three types of DWs can be easily seen from the contribution of wave 2 to the total E-P flux and its convergence. In the pre-SSW periods, wave 2 cooperate with wave 1 to enhance the upward propagation of planetary waves and therefore the anomalous E-P flux convergence for BOTH and EA (Fig. 10a, b). Although the wave 2 is enhanced to propagate upward to the stratosphere for NA and NDW, the E-P flux is little converged, and the wave driven change for the circulation is not present (Fig. 10c, d).

This difference in the wave 2 forcing also exists in the post-SSW period for the three types of DWs (Fig. 10e-g). The vertical component of EP flux anomalies by wave 2 reverse the sign for BOTH and NA (Fig. 10e, g), while the upward propagation of wave 2 is still present for EA (Fig. 10f). A a consequence, the wave 2 forcing for the weakening of the polar vortex still persists after the SSW onset for EA, while this forcing for BOTH and NA types by wave 2 has terminated. Changes in the EP flux anomalies and the divergence anomalies by wave 2 are not noticeable for NDWs (Fig. 10h).

The wave 1 and wave 2 can exert different effects on the stratospheric polar vortex (Baldwin et al., 2021; Lindgren and Sheshadri, 2020; Yang et al., 2023). The former propels the polar vortex displaced away from the Arctic, while the latter elongate and splits the vortex. It can be seen that the ratio of displacement events to split events is 7/6 for BOTH events, 6/8 for EA events, 4/2 for NA events, and 12/7 for NDW events. Previous studies (Anstey et al., 2013; Kidston et al., 2015; Lehtonen and Karpechko, 2016) found that 2–3 weeks before (after) the displacement SSW, it is anomalously warmer (cooler) in the



southeastern US and colder (warmer) in Eurasia, while in the month before (after) the split SSW, the probability of both continents (North America and Eurasia) being cold at the same time is high. The different percentage of the displacement or split SSWs might further account for the different circulation and t2m anomalies for different types of DWs.

**5 Conclusions**

SSWs show strong inter-case variability for their impact on the troposphere and near surface. In this paper, SSWs are classified into different groups based on whether there is downward impact and on where the downward impact occurs. Using a newly proposed method, this study further classifies the observed 52 SSWs into four groups: DWs with near surface impact over both Eurasia and North America

(BOTH), DWs with impact over Eurasia (EA), DWs with impact over North America (NA), and NDWs. Finally, 33 DWs are selected from the observed 52 SSWs, compared with the 19 NDWs. Among 33 DWs, three are 13 BOTH events, 14 EA events, and 6 NA events. To well distinguish the potential impact diversity of SSWs, the tropospheric circulation response and the near surface behaviors following these four types of events are revisited in this study. The main finding in this study are as follows and

summarized in Fig. 11.

1) The mean intensity of NDW events in terms of the NAM and the circulation anomalies in the stratosphere and troposphere are nearly half weaker smaller than DWs events. Comparing the downward impact of three types of DWs, the persistency of the negative NAM pattern varies with the DW type. On average, the negative NAM signal in the lower stratosphere can last for >60 days for BOTH and EA,

while it only lasts for ~40 days for NA events affect the troposphere for only 40 days. Further, the dipping pattern at the near surface exhibits a continuous negative NAM signal for BOTH and EA, while it is replaced by positive NAM at both the beginning and end of the SSW for NA.

2) An isentropic vorticity analysis reveals that anomalously high PV air can move southwards and downwards from the polar stratosphere and enter the troposphere in mid- and high latitudes, which can

well track the movement and source of anomalously cold air. The cold anomalies over Eurasia and over North America has precursors in the upstream ocean regions before the SSW onset. Anomalously low PV appears over North Atlantic 10-30 days before the EA onset and anomalous high PV forms over North Atlantic 0-30 days afterward. In contrast, anomalously high PV appears 10-20 days before the NA onset to 0-10 day afterward over North Pacific – Alaska. The PV anomaly evolutions for BOTH has a joint



characteristic of both EA and NA. Namely, anomalously high PV forms over North Atlantic and North Pacific – Alaska as the anomalously low PV air enters the Arctic before the SSW onset for BOTH, while anomalously high PV air sweeps the continents soon afterward. In contrast, the t2m and 315-K PV anomalies are weaker and more scattered.

3) Although the impact of DWs and the NDW on the near surface temperature is contrastingly different,

such a difference is very minor for the total precipitation anomalies. Following the onset of all types of SSWs, the rainfall band show a southward shift especially over the Atlantic–Europe sector, exhibiting a rainfall anomaly dipole in the Western Hemisphere. Namely, the precipitation in subtropical Atlantic – Mediterranean Sea increases, while precipitation is high-latitude rainfall decreases. Previous studies have found that the southward shift of the precipitation band is associated with an equatorward shift in the

storm track, and the intensification of moving cyclones at low latitudes (e.g., Thompson and Wallace, 2001; Huang and Xie, 2015).

4) The dynamical processes during DWs and the NDW are various before and after the SSW onset. In the pre-onset period, the negative NAM has been shaped at surface for BOTH and EA, while weak NAM still dominates at surface for NA, although anomalous high begins to form over the Arctic. In the post-

onset period, the anomalously low band in midlatitudes show different structures for DWs. Three low centers are clearly present over Western Europe, East Asia, and northern North America especially in the lower stratosphere for BOTH, while only the low center over East Asia still exists for EA and NA. The near surface NAM pattern is well organized with the midlatitude negative pressure anomalies over sea larger than over lands for BOTH and EA, while the pressure anomalies are larger over lands than over

oceans for NA. For the NDW, the circulation anomaly amplitude is only half or even one third of that for DWs.

5) The dynamic differences for the three types of DWs are also featured by the wave activities. Firstly, the wave-1 forcing for them (BOTH, EA, and NA) shows a similar structure in the pre-onset period: the upward propagation of wave 1 increases, and the anomalous E-P flux convergence denotes a dissipation

of wave 1 in the lower stratosphere. In the post-onset period, the enhancement of upward-travelling wave 1 terminates for BOTH and EA, although a small enhancement is still present for NA. Secondly, the wave 2 forcing behaves in very different spatiotemporal structures: the wave 2 is enhanced to propagate upward in the pre-onset period for all types of DWs, but the wave dissipation in the stratosphere is only detected for BOTH and EA. In the post-onset period, the upward propagation of wave 2 is instantly suppressed



for BOTH and NA, while the wave 2 forcing is still strong for EA.

6) Considering that displacement and split events have different impacts on the two continents (Anstey et al., 2013; Kidston et al., 2015; Lehtonen and Karpechko, 2016), a larger proportion of displacement SSW for NDWs and NA might explain their weak downward impact on near surface especially over Eurasia.

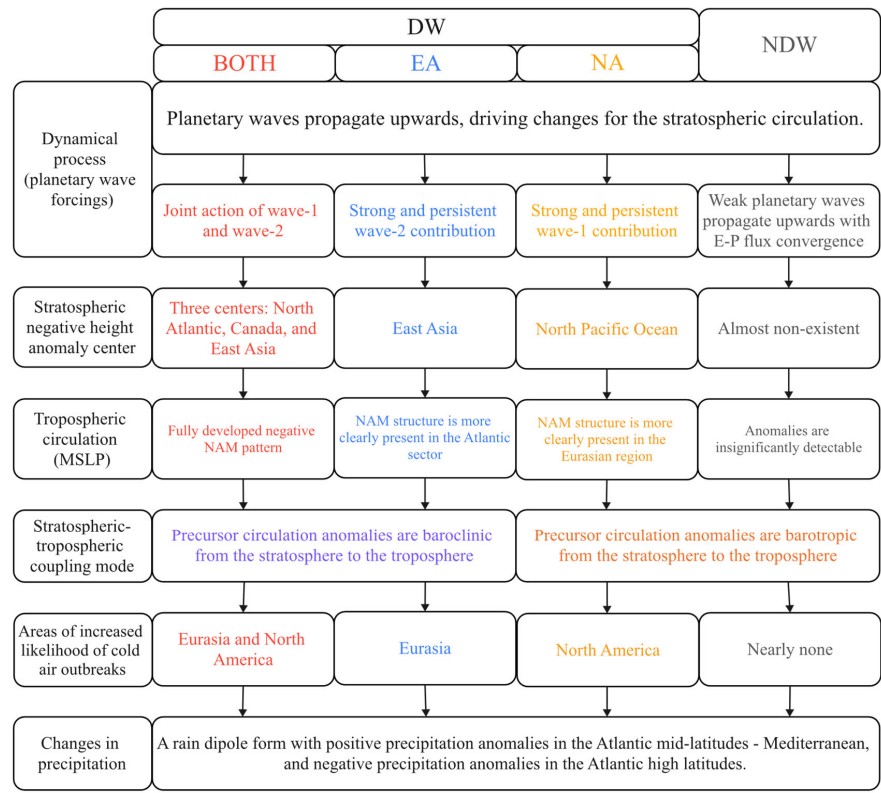

**Figure 11.** Schematic charts comparing the three types of DWs and the NDW. Aspects of comparison include preceding wave forcings by wave 1 and 2, the circulation pattern in the stratosphere, the stratosphere-troposphere coupling, and the downward impact on near surface.

Compared with previous studies (Anstey et al., 2013; Domeisen et al., 2020), our study reveals the diversity of the DW events and distinguish the potential impact on both continents in the Northern Hemisphere. The findings underscore the significant regional diversity in the tropospheric response to different SSW types, with distinct durations, intensities, and spatial patterns. These new insights enhance our understanding of the mechanisms driving surface impacts and could improve our understanding of

weather and climate variability associated with SSWs. This consideration is reasonable, because cold air



outbreaks on both continents are not in pace most of the time (Butler et al., 2017; Yu et al., 2024). However, the diverse rainfall anomaly patterns for DWs and the NDW are insignificant likely due to very limited samples for DWs. Using more samples from model outputs, a deeper understanding of different types of DW events is possible, left for future investigation. Further, the distribution of various types of

events (see Table 1) shows strong interdecadal variability in past decades. Whether this change is an internal climate variability or forced by global warming due to anthropogenic emission is still unknown, worth exploring in the future.

**Data availability**

The ERA5 reanalysis is available from the ECMWF
(https://cds.climate.copernicus.eu/cdsapp#!/dataset/reanalysis-era5-pressure-levels-monthly-means?tab=form).

**Author contributions**

RL and JR designed this research. RL and JR analyzed the data. RL provided the data analysis methods. RL wrote the manuscript draft. JR reviewed and edited the manuscript.

**Competing interests**

The contact author has declared that none of the authors has any competing interests.

**Acknowledgments**

The authors express their gratitude to the National Natural Science Foundation of China for the funding support. The authors thank the High Performance Computing Center of Nanjing University of
Information Science & Technology for their support of this work. The ECMWF is acknowledged by the authors for the accessible modern reanalysis data.

**Funding**

This research was supported by the National Natural Science Foundation of China (Grant nos. 42361144843, 42322503, and 42175069) and the Qinglan project of Jiangsu.



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
