# Peer review of "Flavor identification of the stratospheric sudden warmings based on the downward tropospheric influence"

_EGUsphere, 2024_

## Author Comment (AC1)

**Response to Reviewer # 1**

Review for "Flavor identification of the stratospheric sudden warmings based on the downward tropospheric influence" by Lu and Rao.

Summary
The authors classify sudden stratospheric warmings (SSWs) into those that propagate downward (DW) and those that don't (NDW), and then further classify the DW events into those with cold air outbreaks over Eurasia (EA), North America (NA), and BOTH. They look at the dynamical differences before and after each type of event.

Overall
There are potentially interesting findings here, and the presentation/figure quality is very good, but overall it is difficult to argue that the results are very meaningful when most of it is a result of how the authors constructed the analysis. In addition, there simply aren't enough samples in the reanalysis record (52) to divide further into 4 categories and expect statistically meaningful or interpretable results. Finally, the article is greatly hindered by a lack of proof-reading and grammar issues. Overall, I can't recommend publication at this time.

Response: Thank you for your precious time and effort in reviewing our manuscript. We well double-checked our paper this time, and the language and grammar issues were improved. Further, we tried to explain the significance of this study and explored the near surface climate anomalies associated with the NAM propagation.

The biggest issue for our study is the sample size, especially for the NA type. Actually, the sample size exceeds 10 for most types. To well explain the result and to reduce the uncertainty due to the low sample size, all figures are shown with the significance level overlaid over the composite anomalies. We also inserted more discussion this time for the sample size, and the results can also be further verified with model simulations that have more samples in the future study. Please see our response to each of your comments below.

1. Sample size. I think it is hard to make any substantial statistical arguments with 33 DW events and 19 NDW events. Note that White et al. 2019 find that at least 35 events must be present to get robust differences between DW and NDW events (https://journals.ametsoc.org/view/journals/clim/32/1/jcli-d-18-0053.1.xml). The 33 DW events are then even further divided into EA, NA, and BOTH surface impacts; for NA, there are only 5 samples, according to Table 1. This makes the significance levels for NA almost meaningless. Moreover, most of the paper discusses differences between EA, NA, and BOTH, but no statistical tests are applied to actually quantify the differences. My feeling is that almost none of the noted differences in these categories are likely to be statistically different, because the sample size is simply too small. This is why using a model to get many thousands of SSWs would be useful.

Response: This article only uses ERA5 reanalysis to compare the possible impact of NDW and DW. The sample size is relatively small for NA events. However, the

composite anomalies for the NDW and each type of NW is significant (see Figures 1, 3, 4, 5, 6, 7). The sample size really can impact the robustness of the composite results, but an increase of the sample size might not change the anomaly pattern.

To well address your concern, we made several revisions this time. Firstly, we discuss the possible issue resulting from the limited sample size in reanalysis. Secondly, we provide an insight into the future study that use more sample size from CMIP5/6 models to verify our research. Namely, we expected to increase the sample size and validate the conclusions using model data. However, due to the limited scope of this study, we did not insert too many figures. We can show the figure exclusively for your reference (Fig. R1).

[Figure]

**Fig. R1.** Same as in Figure 1, but for model data. CESM2-WACCM historical simulations (r1i1p1f1 to r3i1p1f1) were used for composite (the sample size for each type exceeds 35).

Revision related to this comment can be found at Lines 21-22, L546-548, etc.

2. Use of stratospheric reanalysis prior to 1958. This is just model data at that point, not reality- in which case, why not use model data to up your statistics? If model data simply is not available (which it should be- many high-top CMIP pi-Control runs could likely simulate downward coupling very well), at least don't use reanalysis before 1958, as any SSWs prior to then are almost certainly going to be in model land anyway and not strongly constrained by observations.

Response: We used the stratospheric reanalysis prior to 1958 to increase the sample size, which is more reliable than the model data at least. Models show discrepancy in the SSW simulations. To well address your concern, we insert a sentence this time: "The data prior to 1958 are used with an expectation of selecting more SSW samples." (L99-100).

This comment is very similar to the first one. We insert in the discussion about the limitation of our study: "White et al. (2019) found that at least 35 events should be present to get robust differences between DW and NDW events. Using more samples from model outputs, a deeper understanding of different types of DW events is possible, left for future investigation." (L546-548)

3. A huge part of the analysis section discusses differences between EA, NA, BOTH DWs and NDWs without seeming to acknowledge/recognize that most of the differences in the post SSW period are there *by construction* of how the authors classified these events. If you classify NA events by cold air temperatures over NA, they are going to have cold air temperatures over NA, and the corresponding large-scale meteorology to give you those cold air temperatures. Additionally, there are several statements about how NDWs are "less organized" and don't show similar spatial structures; but this isn't a fair comparison, because the NDWs *aren't* divided up by regional surface temperature responses and instead average together all the variability. A more apples-to-apples comparison would be if you also divided the NDWs up in the same way (but again, this would cause very small samples). Overall, I don't think much is learned except in the pre-SSW period since that is somewhat more independent from the classification method itself.
Response: Thank you for your comments. We agree with you that the surface anomaly pattern for SSWs is somewhat dependent on the pre-SSW tropospheric signals. We did not deny the important role of the tropospheric circulation before SSW onset.

Firstly, we classified SSW events based on surface characteristics using the NAM index, consistent with previous studies (e.g., Natarajan et al. 2019; White et al. 2019). Secondly, we clarified the NDs according to the areas where cold anomalies developed.

Please refer to Fig. 3, and you might see that the t2m anomaly amplitude over the land is larger than over oceans. Therefore, for NDs, the t2m anomaly center is mainly situated over the land. Therefore, a comparison between both lands is meaningful. Even if we averaged all DW events (BOTH, EA, and NA), the significant cold anomalies are still present over Eurasia and North America. It is more meaningful to classify the DWs into BOTH, EA, and NA than for NDWs. Since NDWs mainly refer the events that do not show downward impact on the near surface, we did not see a necessity of further classifying the events into several types. We agree that some NDWs might be accompanied with cold anomalies over Eurasia or North America that are caused by tropospheric variability or other external forcing, but the composite shows no significant signals over land (Fig. 3). Namely, we doubt about the necessity of further classifying the NDWs into different types. In addition, the case number is also too limited, and a robust result can not be possible. After careful consideration, we did not classify NDWs.

For more details, please see the method section. (L108-161)

4. Text needs to be edited for grammar and just general proof-reading. I started off adding suggestions for Technical Edits, but to be honest, could not keep up with the level of errors, and so at some point I stopped adding to the Technical edit list below. In many cases, the edits go beyond a language barrier and just seem to be due to a lack of reading the paper carefully before submitting (e.g., sentences are repeated twice, results are explained for different regions than are stated, etc).

Response: We apologize for the many grammar or word errors in the text, which was due to the frequent editing by the two authors before submission. We double-checked the manuscript this time, and all the edit errors and grammar problems were corrected this time.

5. Line 5, 33, and throughout: Suggest using "sudden stratospheric warming" phrasing instead to be consistent with most recent literature (Baldwin et al. 2021; see also, Butler et al. 2015, BAMS, for historical context in using this phrasing).

Response: We have changed all the expressions in the text from "stratospheric sudden warming' to 'sudden stratospheric warming". The review paper from Baldwin et al. 2021 is also cited. (L5, 34, 44, 67)

6. Line 20, 22: Not clear what is meant by "deflection of the anomalous high". Do you mean "shift"?

Response: Revised. (L23)

7. Line 22, 26: If there are only 33 DW events, are they further divided into type of SSW? This seems like the statistics would be very poor. See also, Major Comment #1.

Response: The sample from the ERA5 reanalysis is indeed much less than long outputs from models. However, at the present time we tried to reveal the fact that is observed in the reanalysis. The limited samples might give a weak significance level, but composite pattern is still significant.

To well address your concern, we remind readers of the limited sample size: "Based on the limited samples from ERA5, the shape of the anomalous polar high varies with the DW type ⋯" (L21-23)

8. Line 24: Isn't it somewhat by construction that the NDWs have a weaker impact on the troposphere?

Response: The DW and NDW was classified by the NAM index, rather than a pure construction. We clearly describe the procedure in the method section. Please refer to the method section for more details.

9. Line 28: Is it really showing robust diversity across events, or just the large contribution of sampling variability?

Response: It is mainly due to the diversity across events between NDWs and DWs, and among BOTH, EA, and NA events. Note that we show the significance level for the composite in dots. Sampling variability does not produce so high a significance level.

10. Line 38: Butler et al (2015) was about defining SSWs, and did not quantify drivers of SSWs. A more appropriate reference here might be Polvani and Waugh (2004).
Response: Revised. (L40)

11. Line 44-45: I'm not sure what is trying to be said here, but this is not true. Stratosphere-troposphere coupling is a two-way process, e.g., there are certain tropospheric patterns that precede stratospheric variability, which then couples back to the surface. Maybe instead "Stratospheric variability associated with SSWs affects the troposphere through stratosphere-troposphere coupling processes" ? Alternatively, just remove this sentence and start the paragraph with "Several mechanisms⋯"
Response: Revised as suggested: "Stratospheric variability associated with SSWs affects the troposphere through stratosphere-troposphere coupling processes (Hitchcock and Simpson, 2014; Wu and Reichler, 2019)". (L45-46)

12. Line 53-55: Yes, but this can really only explain the downward descent of the anomalies to the troposphere. There are aspects of wave-mean flow interaction that cannot fully explain the amplified response at the surface. It's been proposed that the eddy feedbacks in the troposphere must play some role, but this is not well understood. For a review, see Kidston et al. 2015
(https://www.nature.com/articles/ngeo2424)
Response: We added some discussion this time:
"As a consequence, the stratospheric disturbances cause significant downward impact on tropospheric circulation changes and near surface climate anomalies (Colucci and Kelleher, 2015; Dall'Amico et al., 2010), which are proposed to be amplified by tropospheric eddy feedback (Kidston et al., 2015)." (L53-56)

Further, Kidston et al. 2015 was also cited in other places:
"The negative phase of the NAM is often accompanied by equatorward shifts in storm paths and tropospheric jets (Kidston et al., 2015) ⋯" (L61-62)
"Previous studies (Anstey et al., 2013; Kidston et al., 2015; Lehtonen and Karpechko, 2016) found that 2–3 weeks before the displacement SSW, it is anomalously warmer in the southeastern US and colder in Eurasia⋯" (L461-)
"Considering that displacement and split events have different impacts on the two continents (Anstey et al., 2013; Kidston et al., 2015; Lehtonen and Karpechko, 2016), ⋯" (L523-)

13. Line 60: This needs to specify "The negative phase of the NAM is", not just "Changes in the NAM are", since what is discussed thereafter are specific to one phase.
Response: Revised. (L61)

14. Line 68-69: Is this really true? Or is it just the large internal variability present during any individual SSW event that dominates how much influence the SSW will

be able to have on the surface? The dynamics of SSWs are by and large very similar across events- so I'm not sure I agree that the differences between SSWs lead to differences in their impacts. For example, see Maycock and Hitchcock 2015- when you have a lot of simulated SSWs, differences in their surface impacts arise largely from sampling, not because of inherent differences in the SSWs (https://agupubs.onlinelibrary.wiley.com/doi/10.1002/2015GL066754).

Response: We agree that when the sample increase, the difference between SSWs might compromise. Large model evidence shows that as SSW sample increases, the across-event difference gradually weakens. However, this does not contradict with the diversity of observed SSWs. The SSWs in reanalysis show every large difference, and the observations are more complex than models. If we examine the SSW case by case in those years, their difference is very clearly present.

To well address your concern, we insert discussion this time: "It is the differences between individual SSWs and the background conditions in observations that distinguish their influence on the troposphere. As a consequence, the tropospheric response signals vary in extent, area and scope." (L69-71)

15. Line 85: Does there need to be? Given the huge range of possible internal variability, I think it would be almost impossible to cleanly classify by surface impact.

Response: We agree with you that we tried to extract the signals from large internal variability. If the composite shows a high significance level, we still have an excuse to believe that the difference between DWs really exists. For most cases, the signals are really much smaller than chaos. However, it does not signify that the signals can not be extracted from the large variability.

Our study begins from the assumption that the surface impact can help classify the DWs. (L85) The composite results are also present in our study in following parts. Thank you.

16. Line 105: The definition that is described here is that from Charlton and Polvani (2007). The WMO never had a definition that specifically was based on zonal winds alone; see Butler et al. (2015). Likewise, on line 115, this aspect of the definition was developed in Charlton and Polvani (2007), not White et al. (2019).

Response: All were revised. (L110, 120)

17. Line 113-114: Not sure what is meant here by "zonal winds oscillating between westerlies and easterlies." As the corrigendum in CP07 states, the events must be separated by more than 20 days of consecutive westerlies; not just 20 days apart. This number of days arises from roughly double the radiative timescales in the mid-stratosphere, to avoid double counting what might otherwise be the same dynamical event.

Response: We have clarified this sentence and avoid this illegibility:

"Considering the zonal winds change radically even after the SSW onset, the onset date of the two SSW events must be more than 20 days of consecutive westerlies apart (roughly double the radiative timescales in the middle stratosphere)." (L117-119)

18. Line 127: The average NAM is negative at what level? Both? Either?
Response: Clarified: "the average NAM index is negative at both 850 hPa and 150 hPa." (L132)

19. Line 134-5: Provide citations for this statement, or refer to your own Figure.
Response: This sentence gives an assumption: "It is assumed that following the DW event onset, continental cold anomalies can develop over Eurasia and/or North America, implying an increase in cold air outbreaks after the DW SSW onset." (L139-140)

20. Line 147-8, 152: This method was actually from Mitchell et al. 2011, and then was adapted by Seviour et al. 2013: Seviour, W. J. M., D. M. Mitchell, and L. J. Gray (2013), A practical method to identify displaced and split stratospheric polar vortex events, Geophys. Res. Lett., 40, 5268–5273
Response: Thank you for your suggestion. We have made the necessary changes. "The classification of SSWS into split and displacement events is based on the method initially developed by Mitchell et al. (2011) and adapted by Seviour et al. (2013) using the two-dimensional (2D) moment analysis method." (L152-154)
"We choose the thresholds as the most equatorward 5.7% of centroid latitudes and largest 5.2% of aspect ratios, yielding a threshold of 62.9° N for centroid latitude and 2.46 for aspect ratio, respectively (e.g., Mitchell et al., 2011; Seviour et al., 2013)." (L158-161)

21. Line 164: There are a lot of other undefined variables here.
Response: Added. "…f is the Coriolis parameter, $\theta$ is the potential temperature, $g$ is gravity acceleration, $p$ is air pressure, $\vec{k}$ is the vertical unit vector, and $\vec{V}$ is horizontal wind vector at isentropic surface." (L170-171)

22. Line 190-215: Are any of these differences across DWs significant though? My guess is no, but that should be assessed here (the significance of the individual composite anomalies is assessed, but this is different than stating that the composites themselves are different and then testing that- you need to do a difference in means test, using the sample size/standard deviation of each composite group). Given that NA only has about 5 samples, it's going to be hard to judge whether you're not just seeing sampling variability.
Response: We consider your suggestion, and insert two typical differences for the paper. One is the difference between BOTH and NDW, and one is the difference between EA and NA. Those two differences are mentioned times and should be emphasized in the paper. We show those two differences for a simplicity and avoid much too wordy description. Please refer to the revised Fig. 1 for more details.

Places related to this issue is listed as follows.

"…while it is very short in the persistent time and show a low significance level for NA." (L217-218)

"The difference between BOTH and NDW and between NA and EA is most significant in the troposphere and near surface, implying a diversity in the persistency and NAM intensity among SSW types." (L221-223)

23. Line 211: By requirement, it has to have negative NAM850 at least 50% of the time, no matter what type of DW, right? So not sure what is meant here about "short in the persistent time" for NA. Also, I bet if you randomly resample EA and BOTH for 5 samples to equal the sample size of NA, you can get composites that look like NA, just by chance.

Response: We consider your suggestion and tried to verify your assumption with model outputs. Please refer to R1 for more details. The model evidence also shows that the persistency is indeed different for EA and NA. This sentence is not problematic. (L217-218)

24. Line 214-215: This is by construction though, no?

Response: Figure 1 was updated.

25. Line 215: specify "over the North Atlantic"

Response: We have indicated "over the North Atlantic" (L230)

26. Figure 2, line 224-25: I'm worried about how smooth these look. For the blue curve (NA) there are five samples being put in to make this very smooth looking distribution. (this assumes these are 60-day averages, not all 60 days being put into the PDF⋯ not totally clear from the caption).

Response: The NAO index data used for the plot represents the 60-day average following each event. Further, we applied Kernel Density Estimation (KDE) to the index. This method allows the index, which was originally a set of sample data, to be transformed into a continuous probability density function by estimating the data distribution. Caption was updated. (L233, 247-)

27. Line 226-228: I'm not convinced these mean values (or the PDFs in general) are statistically different, particularly for the DWs. You could do a Kolmogorov-Smirnov test to see if the PDFs are separable. Moreover, since the NAO is highly correlated to the NAM, this result (at least the difference between NDW and DW) is also somewhat by construction. Also the values cited here say "NAM" but the NAO is being shown in Figure 2 so I assume that's what is meant?

Response: Following your suggestion, we used the Kolmogorov Smirnov test to examine the difference between PDFs. When the null hypothesis is "two samples from same distributions", the calculated p-value is: 1) BOTH and EA: 0.05; 2) EA and NA: 0.3; 3) BOTH and NA: 0.1.

Further, the typo was also corrected. (L241-242, 246)

28. Figure 3: would be beneficial to add some measure of tropospheric circulation to these figures, such as Z500. That would also help tie into Figure 1.
Response: Thank you for your suggestion. We have added Z500 to Figure 1 and provided further analysis.
"For BOTH and EA, anomalous high appear over the Arctic at 500 hPa, while for NA two anomalous high center appear over North America and Europe, respectively (Fig. 3a-c). The tropospheric circulation anomalies are much weaker for NDWs than DWs (Fig. 3d)." (L259-263)
"The positive height anomaly center for the type BOTH is located over Greenland and the Bering Strait." (L267-268)
"The positive height anomaly center remains over the Arctic Ocean and Canada." (L270-271)
"The positive height anomaly centers move from Canada and Europe to Greenland and Central Asia, respectively." (L273-274)
"For the NDW, the t2m and 500-hPa height anomalies over both lands are very scattered and less organized, although patches of warm anomalies are observed over Asia (Fig. 3h)." (L274-275)

29. Line 239-255: It should be re-emphasized throughout that the cold anomalies over the region of definition (e.g., NA, EA) are by construction, so the fact that these regions show different anomalies is not unexpected nor does it tell you much about why they are different in other regions.
Response: We remined of readers of this construction at the beginning of this paragraph: "In the post-SSW period, the t2m anomaly pattern is contrastingly different for the three types of DWs and the NDW, consistent with the construction for t2m anomalies." (L263-264)

30. Line 247, 287-88: This isn't a fair comparison. Here, you've divided the DW into categories that are defined based on persistent cold events in a given region, so they're going to look very distinct and organized—by construction-- compared to the NDW events which are thrown into a pool of all types of surface impacts. A more fair comparison would be to divide the NDW events using the same classifications used in the DW events based on regional cold air outbreaks, and compare those groups…
Response: In our study, we are more concerned about the SSWs that show the downward impact on the troposphere. It has the possibility that the t2m over Eurasia or North America show cold anomalies. However, our study is more concerned with the cold anomalies associated with the SSW. In the NDWs, we have hypothesized that SSWs does not affect the troposphere and the near surface. The cold anomalies for some NDWs over Eurasia or North America are not associated with the SSW events. Therefore, it is more logical not to classify the NDWs any more. Another reason of not classifying the DWs is the too limited sample size.

31. Figure 4: would be easier to read if instead of I-IV the regions (Europe, Asia, etc) were provided in the label on the plot

Response: Thank you for your suggestion, we have made the necessary modifications.

32. Line 256: ? There is no temperature anomaly reversal over Europe for NA in Figure 4.

Response: Revised "For the type NA, the reversal of temperature anomaly sign is observed over North America (US and Canada)." (L288-289)

33. In many cases, the highest IPV is associated with the Greenland block (e.g. BOTH and NA) yet the authors are focused on much smaller positive anomalies over the N. Pacific as a source for the NA cold. Yet it's been shown in the literature that the Greenland blocking pattern (also associated with the negative NAM/NAO) would advect cold air over at least eastern North America. To me this seems to suggest this is the dominant forcing of cold over NA.

Response: Thank you for your suggestion. We conducted more analysis on the Greenland region.

"Significant negative IPV anomalies develop over the Greenland region, consistent with the local anomalous high (see Fig. 3a, e), which help advect cold air to eastern North America." (L312-314)

"Similar to BOTH events, the blocking pattern denoted as a wide range of negative IPV anomalies over Greenland is observed, and the upstream winds can advect cold air to eastern North America" (L327-329)

34. Line 304-305: kind of? The pattern is shifted much further poleward for NDW, and looks less spatially coherent.

Response: Thank you for your suggestion, we have made the necessary modifications. "Although the NAM fails to descend to the near surface, the rainfall dipole is biased farther northward for NDW (Fig. 6d)." (L352-353)

35. Line 325: NA looks more baroclinic to me than EA?

Response: We agree and revise this sentence. "Comparing the three types of DWs, the precursor circulation anomalies for BOTH are more baroclinic from the lower stratosphere to the troposphere (Fig. 7a, b), while for NA and EA, the circulation anomalies are nearly barotropic in high latitudes (Fig. 7b, c). The height anoamlies at 100 hPa are much weaker for NDWs (Fig. 7d) than for DWs." (L376-379)

36. Line 330: Again, the majority of post-SSW impacts at 100 hPa are going to be reflecting the event definition…

Response: DWs are classified using the t2m anomalies, and the NAM at 100 hPa is consistently used to select DWs. This sentence is not problematic. (L383-384)

37. Line 335-345: Throughout this section it's often not clear which level is being referred to, 100 hPa or SLP, so make sure it's evident in each instance.
Response: Thank you for your suggestion. We have made the necessary modifications. (L386, 389, 397)

38. Line 352-353: This doesn't seem apparent from Figure 8. At least around 100 hPa, NA seems to have the strongest convergence (blue shading)
Response: We have provided a more precise explanation. The description has been changed to: "It is revealed that the E-P flux convergence anomalies develop in the entire stratosphere for the type BOTH, while for EA and NA, the convergence anomalies are more concentrated in the 50-150 hPa range." (L412-414)

39. Line 370-71: I'm not clear what the first part of this sentence has to do with the second part.
Response: We removed this first sentence that can cause a confuse. (L442)

40. Figure 8-10: I'm confused why the EP flux convergence anomalies are so substantial for days 0-10 in Figure 8, yet when decomposed into wave 1 and 2 in Figures 9-10, they basically disappear/are much weaker. Since wave 1 and wave 2 are the primary components to the total wave flux, I would check and make sure there's not an error in Figure 8···
Response: To well compare the wave 1 and wave 2, only the sum of wave 1 and wave 2 is shown in Figure 8 this time. We agree that only planetary waves can propagate upward into the stratosphere. It can be seen from Figure 8 that the amplitude of EP flux decreases a bit due to the removal of shorter waves.

41. Line 395-397: It is very difficult to parse this type of parenthetical sentence. Please separate into two (or possibly 3-4) sentences.
Response: Revised as suggested. "Previous studies (Anstey et al., 2013; Kidston et al., 2015; Lehtonen and Karpechko, 2016) found that 2–3 weeks before the displacement SSW, it is anomalously warmer in the southeastern US and colder in Eurasia. These studies also found 2–3 weeks after the displacement SSW, the southeastern US is anomalously cold while Eurasia is unusually warm. In the month around the split SSW, the probability of both continents (North America and Eurasia) being cold at the same time is high." (L467-472)

42. Line 418-421: This makes it sound like the authors are implying the air is actually coming from the polar stratosphere, which is simply not true!
Response: This sentence was revised and clarified. "An isentropic vorticity analysis reveals that anomalously high PV air can move southwards from the polar stratosphere and enter midlatitudes,..." (L492-493)

43. Line 429: It's different because you defined it to be so! If you classified by precipitation pattern instead, the patterns would look different, *by construction*.

Response: We revised this sentence: "Adopting relative continental cold anomalies as the criterion for classifying SSWs, changes in precipitation anomalies are not so sensitive to the DW type as t2m anomalies." (L502-503)

**Technical Edits**
Line 19: "response" -> "responses"
Response: Revised. (L19)

Line 29: "distinguish"-> "distinguishes"
Response: Revised. (L29)

Line 37: "the stratospheric precondition might play a…" -> "found that stratospheric preconditioning might play a decisive role in inducing the SSW event, by determining the intensity…". Also, can you add a reference for this statement?
Response: Revised. (L37)

Line 39, and throughout: "that SSW is caused" -> use "a" before SSW when used in the singular, or else say "that SSWs are caused" (as mentioned in Major comments, text should be edited throughout for English style)
Response: Revised. (L40)

Line 46: "mechanism" -> "mechanisms"
Response: Revised. (L46)

Line 51: "affect stratospheric mean flows, and changes in stratospheric background flows in turn affect" -> "affect stratospheric mean flows, which in turn affect"
Response: Revised. (L53)

Line 58: "is usually projected onto the negative phases" -> "usually projects onto the negative phase of"
Response: Revised. (L59)

Line 64: remove "populated", not sure what is meant; also double use of "extreme"
Response: Revised. (L65)

Line 69: either "extent" or "degree", not both
Response: Revised. (L71)

Line 82: "the DW SSWs show a dipping NAM signals" -> I think you mean to say "the DW SSWs show dripping NAM signals" (?)
Response: Revised. (L84)

Line 88: change to "although by definition the NAM…"
Response: Revised. (L89-90)

Line 89: need question mark at end instead of period
Response: Revised. (L91)

Line 101: repetitive phrasing of "isobaric levels vertical levels". This is also not a complete sentence (change "ranging" to "range" and then put the surface data part in a separate sentence)
Response: Revised. (L104)

Lines 105-115: Suggest carefully re-reading and editing this section; many grammar issues or sentence structure problems.
Response: Revised. (L110-122)

Line 118: change "extracted" to "calculated"
Response: Revised. (L123)

Line 147: change to "The classification of SSWS into split and displacement events is…"
Response: Revised. (L152)

Line 165: EP-flux acronym is never defined
Response: Revised. (L173)

Line 174: "characterize" -> "characterizes"
Response: Revised. (L181)

Line 179: just say "The NAM index can be used to describe"… Also, can remove "for DWs". It describes the extent of downward propagation whether they are DW or NDW.
Response: Revised. (L186)

Line 182: remove "radical"
Response: Revised. (L188)

Line 194-195: for NA? or what is this sentence referring to?
Response: Revised. (L200-202)

Line 245: "EA" should be "NA". Also I think that western Europe should be "eastern Europe"? (warm anomalies in western Europe seem to weekend compared to before SSW, in 3g)
Response: Revised. (L271)

Figure 6, 8-10: labels for negative values on colorbar are not shown
Response: Revised. (Figure 6, 8-10)

Line 309: change to  "the circulation structure is organized differently for.."
Response: Revised. (L361-362)

Line 320: "the lakes" -> the "Great Lakes"?
Response: Revised. (L372)

Line 337-338: I think I get what you're trying to say but the phrasing here needs improvement
Response: Revised. (L390-391)

Line 344: "even insignificantly detectable" is awkward- also there appears to be stippling over most of the Arctic so not sure what this refers to?
Response: Revised. (L398)

Line 412: just need "weaker" or "smaller" not both
Response: Revised. (L486)

Line 415: sentence repeats itself
Response: Revised. (L489)

Line 466: "pace" -> "place"
Response: Revised. (L544)

---

## Author Comment (AC2)

**Response to Reviewer # 3**

Review of "Flavor identification of the stratospheric sudden warmings based on the downward tropospheric influence" by Lu and Rao.

1. I agree with Reviewer 1 that the statistics are not sufficient to divide the SSWs into four categories. From Table 1, there are 13 BOTH, 14 EA, 6 NA and 19 NDW. Those are very small numbers. In addition, I noticed that 4 of the 6 NA occur in the 2000s, and 6 of the 13 BOTH also occur in the same decade (1970s). There are also many more EA towards the end of the considered time period than early on. Thus, besides the comments from Reviewer 1 that these numbers are not sufficient for statistical analysis and that data before 1958 is questionable, I wonder whether the composites include things like a climate change trend (EA) or potentially decadal variability (BOTH, NA), which both might have nothing to do with SSWs.

Response: When processing data, the long-term trend is removed to minimize the possible impact of climate change trend. We insert the processing procedure this time. "The daily climatology is computed as the long-term mean for each calendar day, and the raw daily climatology is smoothed using 31-day means. The daily anomalies refer to the detrended deviation relative to the smoothed daily climatology." (L105-107)

The three DW types indeed show somewhat preference towards some decades. However, it is still unknown if this preference is related to global change trend or a random coincidence. To well address your concern, we mentioned this possibility. "Further, the distribution of various types of events (see Table 1) shows strong interdecadal variability in past decades. Whether this change is an internal climate variability or forced by global warming due to anthropogenic emission is still unknown, worth exploring in the future." (L548-551)

Limitation of this study is discussed in our revisions. This article only uses ERA5 reanalysis to compare the possible impact of NDW and DW. The sample size is relatively small for NA events. However, the composite anomalies for the NDW and each type of NW is significant (see Figures 1, 3, 4, 5, 6, 7). The sample size really can impact the robustness of the composite results, but an increase of the sample size might not change the anomaly pattern.

To well address your concern, we made several revisions this time. Firstly, we discuss the possible issue resulting from the limited sample size in reanalysis. Secondly, we provide an insight into the future study that use more sample size from CMIP5/6 models to verify our research. Namely, we expected to increase the sample size and validate the conclusions using model data. However, due to the limited scope of this study, we did not insert too many figures. We can show the figure exclusively for your reference (Fig. R1).

[Figure]

**Fig. R1.** Same as in Figure 1, but for model data. CESM2-WACCM historical simulations (r1i1p1f1 to r3i1p1f1) were used for composite (the sample size for each type exceeds 35).

Revision related to this comment can be found at Lines 21-22, L546-548, etc.

2. Furthermore, the cited paper by Jucker (2016) has shown that composite evolution and surface impact can be very different depending on the exact definition of "downward propagation". For instance, they show that simply checking for expected anomalies after SSW onset captures periods of internal variability where these anomalies might already exist before the SSW can influence the surface. Compare this, for instance, to the statement on lines 314-315, "The negative NAM pattern has well developed before the SSW onset for the type BOTH."

Response: We agree that the internal variability in the stratosphere and troposphere is mixed together, which is impossibly disentangled. Strictly speaking, the SSW is also a radical stratospheric variability. Certain tropospheric modes are a signal that often accompanies the development of SSW. As the precursor for SSWs, the tropospheric internal variability is inextricably linked to the onset of SSW. Therefore, it necessitates a study that well compared different types of DWs.

In this study, the pre-existing internal variability is regarded as one of aspects that well distinguish the DW types. Namely, we use surface anomalies as a classification criterion to classify the DW types. We show the 40-day mean, and the memory of troposphere itself can hardly exceed 40 days.

To well address your concern, we revised the method section.
"It is assumed that following the DW event onset, continental cold anomalies can develop over Eurasia and/or North America, implying an increase in cold air outbreaks

after the DW SSW onset. To better describe and distinguish the DWs, we further divide DWs based on the inland temperature anomalies within 40 days after the onset of DWs." (L139-142)

Jucker (2016) is also cited in our paper. Thank you.

3. I am also a bit sceptical about the use of the sign of the NAM as a measure of downward propagation (lines 125-129): The NAM is a zonal mean quantity, but EA and NA are explicitly defined as strongly zonally asymmetric surface anomalies (impact in one region but not another). So why to the authors think using the NAM is the best way to define downward influence of SSWs? Is it not possible that they are missing several occurrences where EA or NA are anomalously cool but the zonal mean is still neutral (and thus this would be classified as NDW)?

I think these are major points concerning the design of the study which need to be re-considered, and I therefore do not recommend publication in the current form.

Response: We use NAM to define DWs and NDWs, but we do not use NAM to define BOTH, EA, and NA. In our study, we are more concerned about the SSWs that show the downward impact on the troposphere. It has the possibility that the t2m over Eurasia or North America show cold anomalies during some NDWs. However, those cold anomalies are not related to the SSW according to the criterion in previous studies and our work.

Our study is more concerned with the cold anomalies associated with the SSW. In the NDWs, we have hypothesized that SSWs does not affect the troposphere and the near surface without downward propagation of NAM signal. The cold anomalies for some NDWs over Eurasia or North America, if they exist, are not associated with the SSW events. Therefore, it is more logical not to classify the NDWs any more. Another reason of not classifying the DWs is the too limited sample size.

---

## Author Comment (AC3)

**Response to Reviewer # 2**

Review for "Flavor identification of the stratospheric sudden warmings based on the downward tropospheric influence" by Lu and Rao.

Summary
Using the reanalysis, this study analyzes the possible impact of the downward-propagating SSWs on the continental climate. To by best knowledge, this is the first study to classify the downward-propagating SSWs into three types: North America, Eurasia, and BOTH. This classification is established on the existing evidence that the composite SSW shows wide coldness anomalies over the Eurasia and/or North America. However, the coldness over both continents is not synchronous for all SSWs. It will improve the understanding of the diversity of the SSW in influencing the near surface. In general, this study is very interesting and worth publishing after a revision.
Response: Thank you for your positive comments.

1. The classification of downward-propagating SSWs is mainly based on the cold anomalies over both continents. Not all of cold temperature anomaly variations are caused by the stratospheric variability, and the composite might filter out the contribution of other variability if the sample size is large enough. Is there any possibility to increase the sample size if the model data are used? For example, CMIP6 provides a large model ensemble dataset, which contains much more samples than ERA5. If those data are used, the stability of the composite results can be well confirmed.

Response: This article only uses ERA5 reanalysis to compare the possible impact of NDW and DW. The sample size is relatively small for NA events. However, the composite anomalies for the NDW and each type of NW is significant (see Figures 1, 3, 4, 5, 6, 7). The sample size really can impact the robustness of the composite results, but an increase of the sample size might not change the anomaly pattern.

To well address your concern, we made several revisions this time. Firstly, we discuss the possible issue resulting from the limited sample size in reanalysis. Secondly, we provide an insight into the future study that use more sample size from CMIP5/6 models to verify our research. Namely, we expected to increase the sample size and validate the conclusions using model data. However, due to the limited scope of this study, we did not insert too many figures. We can show the figure exclusively for your reference (Fig. R1).

[Figure]

**Fig. R1.** Same as in Figure 1, but for model data. CESM2-WACCM historical simulations (r1i1p1f1 to r3i1p1f1) were used for composite (the sample size for each type exceeds 35).

Revision related to this comment can be found at Lines 21-22, L546-548, etc.

2. This study classifies the SSW using the t2m anomalies, which is based on the fact that major SSWs show larger and more significant t2m composite than rainfall composite. Are the zonal band of rainfall anomalies over North Atlantic sensitive to the threshold of downward-propagating SSW type?
Response: Please refer to Fig. 6 in the paper, and we can see that the rainfall anomaly pattern is very similar for the DWs, and the pattern for NDW is farther northward biased. The primary difference lies in the area coverage of significant precipitation anomalies. To well consider your concern, we also verify this conclusion using CMIP6 outputs. Please refer to Fig. R2 for more details. Due to the limited scope of this study, we only show the observational facts in this paper. The model evidence is left for future study.

To well address your concern, we made several revisions.
"Adopting relative continental cold anomalies as the criterion for classifying SSWs, changes in precipitation anomalies are not so sensitive to the DW type as t2m anomalies." (L502-503)
"Using more samples from model outputs, a deeper understanding of different types of DW events is possible, left for future investigation." (L57-548)

[Figure]

**Fig. R2.** Composite rainfall anomalies for BOTH, EA, and NA using model outputs from (a) CESM2-WACCM r1i1p1f1, (b) CESM2-WACCM r2i1p1f1, (c) CESM2-WACCM r3i1p1f1; and (d) all historical simulation members.

**Other comments**
L38: determine => determines; Butler et al., => remove ","
Response: Revised. (L40)

L44: lead => leads
Response: Revised. (L45)

L46: mechanism => mechanisms
Response: Revised. (L46)

L56: and have => remove
Response: Revised. (L57)

L64: populated => growing ; extreme extreme => remove one
Response: Revised. (L65)

L101-102: Reviesed as "The isobaric levels extend from 1000 hPa to 1 hPa, and the horizontal resolution of the data is 0.25° latitude by 0.25° longitude."
Response: Revised. (L104-105)

L105: SSW => SSWs; All => all
Response: Revised. (L110)

L107: The 1 November => 1 November
Response: Revised. (L112)

L111: using => used; do => does
Response: Revised. (116)

L113: twice => more than once
Response: Revised. (L117-119)

L117: ERA => remove
Response: Revised. (L122)

L120: the cosine => cosine
Response: Revised. (L125)

L125: Natarajan et al., => Natarajan et al.
Response: Revised. (L130)

L130: White et al., (2019) => White et al. (2019)
Response: Revised. (L135)

L147, 151: Esler et al., (2009) => Esler et al. (2009)
Response: Revised. (L153, 155)

L174: Characterize => characterizes
Response: Revised. (L181)

L186: return => returns
Response: Revised. (L192)

L188: are present => is present
Response: Revised. (L194)

L198: persist => persists
Response: The original sentence has been modified.

L231: interval => intervals
Response: Revised. (L252)

L232: region => regions
Response: Revised. (L253)

L243: move => moves
Response: Revised. (L269)

L259:stable moderately warm state => moderately warm stable state

Response: Revised. (L292)

L267: Lu and Ding, 2015; => Lu and Ding, 2015
Response: Revised. (L306)

L309: the circulation structured is different => the circulation structure is differently organized
Response: Revised. (L361)

L330: amplitude .. are => amplitude is
Response: Revised. (L383)

L337: cut => cuts
Response: Revised. (L391)

L342: midlatitude => midlatitudes
Response: Revised. (L395)

L364: all for => for all
Response: Revised. (L431)

L385: reverse => reverses
Response: Revised. (L458)

L392: elongate => elongates
Response: Revised. (L465)

L409: finding => findings
Response: Revised. (L483)

L412: weaker smaller => weaker
Response: Revised. (L486)

L421: ocean regions =>oceanic regions
Response: Revised. (L494)

L424: 0-10 day => 0-10 days
Response: Revised. (L497)

L431: show => shows
Response: Revised. (L504)

L440: show => shows
Response: Revised. (L513)